



# Complex refractive indices in the ultraviolet and visible spectral region for highly absorbing non-spherical biomass burning aerosol

Caroline C. Womack[1,2], Katherine M. Manfred[1,2*], Nicholas L. Wagner[1,2], Gabriela Adler[1,2+], Alessandro Franchin[1,2§], Kara D. Lamb[1,2†], Ann M. Middlebrook[2], Joshua P. Schwarz[2], Charles A. Brock[2], Steven S. Brown[2,3], Rebecca A. Washenfelder[2]

[1]Cooperative Institute for Research in Environmental Sciences, University of Colorado, Boulder, CO 80309, USA
[2]Chemical Sciences Laboratory, National Oceanic and Atmospheric Administration, Boulder, CO 80305, USA
[3]Department of Chemistry, University of Colorado, Boulder, CO 80309, USA
*Now at Wolfson Atmospheric Chemistry Laboratories, Department of Chemistry, University of York, York, UK
+Now at Breezometer, Haifa, Israel
§Now at the National Center for Atmospheric Research, Boulder, CO 80305, USA
†Now at Department of Earth and Environmental Engineering, Columbia University, New York, NY 10027, USA

*Correspondence to*: Caroline C. Womack (caroline.womack@noaa.gov)

**Abstract.** Biomass burning aerosol is a major source of PM$_{2.5}$, and significantly affects Earth's radiative budget. The magnitude of its radiative effect is poorly quantified due to uncertainty in the optical properties of aerosol formed from biomass burning. Using a broadband cavity enhanced spectrometer with a recently increased spectral range (360 – 720 nm) coupled to a size-selecting aerosol inlet, we retrieve complex refractive indices of aerosol throughout the near-ultraviolet and visible spectral region. We demonstrate refractive index retrievals for two standard aerosol samples: polystyrene latex spheres and ammonium sulfate. We then retrieve refractive indices for biomass burning aerosol from 13 controlled fires during the 2016 Missoula Fire Science Laboratory Study. We demonstrate that the technique is highly sensitive to the accuracy of the aerosol size distribution method, and find that while we can constrain the optical properties of brown carbon aerosol for many fires, fresh smoke dominated by fractal-like black carbon aerosol presents unique challenges and is not well-represented by Mie theory. For the 13 fires, we show that the accuracy of Mie theory retrievals decreases as the fraction of black carbon mass increases. At 475 nm, the average refractive index is $(1.635 \pm 0.056) + (0.06 \pm 0.12)i$.

## 1 Introduction

Biomass burning is one of the largest global contributors to accumulation mode aerosol mass, with estimated emissions of 15 – 57 Tg yr$^{-1}$ (Pan et al., 2020). In the United States, biomass burning is calculated to contribute 2.4 Tg yr$^{-1}$ of PM$_{2.5}$ aerosol, which accounts for one-third of the primary aerosol source (Watson, 2002; Wiedinmyer et al., 2006). In the western U.S., increased wildfire frequency, wildfire duration, and active fire season have been associated with increased spring and summer temperatures (Westerling et al., 2006; Dennison et al., 2014). Biomass burning aerosol plays an important role in Earth's radiative budget by absorbing and scattering light (Boucher et al., 2013). Biomass burning smoke contains a complex



mixture of particles with varying composition, morphology, size, and refractive index. The two major absorbing components are fractal-like graphitic material (black carbon) and light-absorbing spherical organic aerosol (brown carbon), with smaller contributions from dust (Li et al., 2003; Pósfai et al., 2003; China et al., 2013). One recent study estimated that biomass burning
contributes 27% of black carbon emissions and 62% of primary organic aerosol globally (Wiedinmyer et al., 2011).

The interaction of an aerosol particle with light can be calculated from its size, morphology, mixing state, and complex refractive index (RI). The RI is an intrinsic physical property of the particle, and is described as $m = n + ki$, where $n$ represents the scattering component and $k$ represents the absorbing component (Moosmüller et al., 2009; Moise et al., 2015). Light scattering by spherical particles is well described by Mie theory, which is a set of solutions to Maxwell's equations representing
the scattering of light by a homogeneous sphere with dimensions similar to the wavelength of the radiation (Moosmüller and Arnott, 2009; Moosmüller et al., 2009). Many global models and satellite retrieval algorithms assume atmospheric aerosol particles are predominately spherical, and calculate total aerosol extinction using Mie theory with a small set of constant RIs for different aerosol types (Liao et al., 2003; Levy et al., 2007; Omar et al., 2009; Lamarque et al., 2012). These RIs are often determined experimentally at a few wavelengths (Nakayama et al., 2010; Zhang et al., 2016), and extrapolated to the full solar
spectrum.

Simple assumptions of sphericity and wavelength-independent RI are not accurate for biomass burning particles, which consist of black carbon with complex morphology (Bond et al., 2013) and brown carbon with wavelength-dependent RIs (Washenfelder et al., 2013; Flores et al., 2014b; Laskin et al., 2015; Bluvshtein et al., 2017). The fractal-like agglomerates of highly absorbing black carbon are known to be poorly represented as spheres (Forrest and Witten, 1979), and are better
described by more complex optical equations, such as Rayleigh-Debye-Gans (Sorensen, 2001; Manfred et al., 2018). Accurate measurements of the size distributions, morphology, and wavelength-dependent refractive indices of biomass burning aerosol are important to better model their climate forcing.

One technique to measure wavelength-dependent RIs is broadband cavity enhanced spectroscopy (BBCES), which combines a broadband light source with highly reflective mirrors to measure total light extinction by particles or gas-phase
species at multiple wavelengths simultaneously (Fiedler et al., 2003; Washenfelder et al., 2008; Varma et al., 2013; Zhao et al., 2017). The absorption and scattering of an aerosol population can be determined by making multiple size-selected measurements of extinction and using Mie theory to retrieve the RI (Washenfelder et al., 2013; Flores et al., 2014a; Zhao et al., 2017; He et al., 2018). This method has been used successfully for retrievals of RI in the near-ultraviolet at 360 – 420 nm using LEDs (Washenfelder et al., 2013; Flores et al., 2014a) and for spectral regions as broad as 380 – 650 nm using laser-
driven arc lamps (He et al., 2018; Li et al., 2020). Complex RIs have been reported for standard aerosol samples, such as nigrosin and Suwannee River Fulvic Acid (Washenfelder et al., 2013; Zhao et al., 2017), and for aged organic aerosol (Flores et al., 2014a; Flores et al., 2014b; He et al., 2018; Li et al., 2020). However, these measurements have all been conducted with spherical, homogeneous particles generated in laboratory or chamber experiments. This method has not been previously demonstrated for measurements of complex populations of non-spherical, highly absorbing black and brown carbon that are
representative of biomass burning aerosol.





In this paper, we determined complex refractive indices for biomass burning aerosol across a very broadband region from 360 – 720 nm. We measured wildfire smoke samples from representative fuels and burn conditions at the Missoula Fire Sciences Laboratory (Manfred et al., 2018; Selimovic et al., 2018). We used the broadband extinction measurements and two independent measurements of the aerosol size distribution together with a retrieval algorithm and several theoretical scattering models to determine complex refractive indices. We validate the retrieval algorithm with spherical, monodisperse populations of polystyrene latex spheres and polydisperse populations of ammonium sulfate. We analyze smoke from 13 fires at the Fire Sciences Laboratory, present detailed examples where the retrieval algorithm can and cannot be used to accurately characterize the complex refractive index, and discuss the implications for remote sensing retrievals.

## 2 Experimental

### 2.1 Overview of the Fire Sciences Laboratory 2016 study

The Fire Influence on Regional and Global Environments Experiment (FIREX) 2016 study was conducted at the U.S. Forest Service's Missoula Fire Sciences Laboratory during October – November 2016 (Manfred et al., 2018; Selimovic et al., 2018). The Fire Science Laboratory contains a ~3400 m$^3$ indoor combustion room for controlled burns (McMeeking et al., 2009; Selimovic et al., 2018). The measurements reported here are for "room" burns, where instruments sampled well-mixed smoke from the combustion room. First, dry fuels weighing 0.24 – 4.4 kg were placed on a ceramic fuel bed and ignited by a heating plate to produce a small, controlled burn that lasted several minutes (Selimovic et al., 2018). The smoke became well mixed in the combustion room within 15 – 20 mins, and persisted under steady state, low-light conditions for 3 – 4 h with minimal dilution and wall loss (Stockwell et al., 2014). Fuels were representative of western U.S. wildfires, and included ponderosa pine (*Pinus ponderosa*), lodgepole pine (*Pinus contorta*), Douglas fir (*Pseudotsuga menziesii*), Engelmann spruce (*Picea engelmanii*), subalpine fir (*Abies lasiocarpa*), manzanita (*Arctostaphylos*), chamise (*Adenostoma fasciculatum*), juniper (*Juniperus*), and sage (*Artemisia*) (Selimovic et al., 2018). A shared aerosol inlet was connected to the combustion room, and it provided flow to a size-selected inlet and the BBCES instrument, as described below.

### 2.2 Broadband cavity enhanced spectrometer for aerosol extinction at 360 – 720 nm

The optical system shown in Fig. 1a is a modified version of the two-channel instrument used previously to measure broadband aerosol extinction at 355 – 420 nm with two LEDs (Attwood et al., 2014; Washenfelder et al., 2015). The light source, optical filters, cavity mirrors, and spectrometer grating were replaced to measure aerosol extinction over a very broadband region, covering 360 – 385 nm and 400 – 720 nm.

We used a single laser-driven white light source (EQ-99FC LDLS; Energetiq, Woburn, MA, USA), containing a continuous-wave laser at 974 nm that pumps a Xenon plasma and outputs a broad spectrum from 170 – 2100 nm (Islam et al., 2013). The light source was temperature-controlled to ~20 deg C and purged with N$_2$ to eliminate O$_3$ production (Washenfelder et al., 2016). The light was transmitted by fiber optic to an off-axis parabola with 0.36 numerical aperture (RC04SMA-F01;



Thorlabs, Newton, MA, USA) for collimation and coupling into the optical cavities. Two colored glass filters (WG345 and WG320) were used to remove UV wavelengths shorter than 345 nm. A dichroic longpass filter with a 400 nm cutpoint (69-897; Edmund Optics, Barrington, NJ, USA) divided the light before it was passively coupled into the two optical cavities.

Additional colored glass filters and a custom interference filter (MLD Technologies LLC, Mountain View, CA, USA) eliminated out-of-band wavelengths.

The optical cavities consisted of two pairs of 2.5 cm diameter, 1 m radius curvature plano-concave mirrors with a high reflectivity coating covering 360 – 385 nm (Advanced Thin Films, Boulder, CO, USA) and 400 – 720 nm (MLD Technologies LLC, Mountain View, CA, USA). The very broadband reflectivity of the visible mirrors was achieved with

multiple layers of thin film coatings applied by ion beam sputtering to the super-polished substrate. Although the visible mirrors span a very broad range with high reflectivity (0.9993 – 0.9999), their losses vary strongly as a function of wavelength (Fig. S1) (He et al., 2018), due to the coating properties and possible contributions from surface roughness, interface roughness, and internal defects. The mirrors were mounted in stainless steel mounts, at either end of a 101.5 cm long aluminum flow cell (2.21 cm ID) with inlets for the aerosol flow and mirror purge flows. The broadband light exiting each cavity was collected by

an F/3.1 lens, coupled into a bifurcated fiber optic bundle, and imaged linearly onto the entrance slit of a grating spectrometer (IsoPlane-160; Princeton Instruments, Trenton, NJ, USA). The spectrometer contained a 300 g/mm grating with 300-nm blaze (centered at 507 nm with 290 – 724 nm bandwidth) and a 16-bit back-illuminated 2048 × 512 pixel CCD detector (PIXIS 2kBUV; Princeton Instruments, Trenton, NJ, USA). During data acquisition, individual spectra were acquired with 0.3 s integration time using the physical shutter of the IsoPlane spectrometer. The CCD readout time was 0.57 s, and so a full

spectrum was taken every 0.87 s. Narrow emission lines from a Hg lamp were measured multiple times each day to determine the wavelength calibration and spectral resolution (average full-width at half-maximum resolution of 1.4 nm between 360 – 720 nm).

### 2.3 Cavity ringdown spectrometer for aerosol extinction at 403 nm

A cavity ringdown spectrometer at 403 nm provided an extinction measurement between the two BBCES channels

at 360 – 385 nm and 400 – 720 nm, and was used as an independent validation of the BBCES extinction. The CRDS method has been described previously (Fuchs et al., 2009) and was not modified from Washenfelder et al. (2013).

### 2.4 Automated flow system

### 2.4.1 Aerosol size selection and size characterization

The custom automated inlet for aerosol size selection is shown in Fig. 1b. The materials and geometry of the flow

system were chosen to maximize particle transmission and minimize inertial losses. The BBCES provides a direct measurement of wavelength-dependent aerosol extinction, but retrievals of complex RIs require extinction measurements of two or more size-selected aerosol populations with consistent composition (Washenfelder et al., 2013). Particle size-selection





was achieved with a custom-built differential mobility analyser (DMA; now available from Brechtel Manufacturing Inc, Hayward, CA, USA (Knutson and Whitby, 1975)). In these experiments, the sample flow through the DMA was 0.5 or 1 vlpm, with a sheath:sample flow ratio of 10:1 or 5:1.

As shown in Fig. 1b, the concentration and size distribution of the size-selected aerosol were determined using two separate methods. The first method was a condensation particle counter (CPC; 3022A, TSI Inc., Shoreview, MN, USA), which sampled at a flow rate of 0.3 vlpm, and measured the total number of particles with a lower aerosol size cutoff of 7 nm. At regular intervals, the DMA and CPC were operated as a scanning mobility particle sizer (SMPS) to determine the aerosol size distribution by scanning the DMA column voltage from 0 – 5000 V, and applying an inversion algorithm (Twomey, 1975; Markowski, 1987). An instrument transfer function was then used to calculate the particle size distribution for each DMA setpoint. The inversion algorithm for the SMPS size distribution is most accurate for spherical aerosol with well-known mobility diameters. The second measurement of the particle size distribution was made using an optical particle counter. The optical particle counter (OPC; LAS 3340, TSI Inc., Shoreview, MN, USA) detects light scattered by individual particles intercepting a 633 nm laser beam, and reports the signal for each particle in 100 logarithmically-spaced diameter bins. A total flow of 0.06 vlpm was used. During laboratory tests after the field campaign, we used a similar optical particle counter with a 1054 nm laser (UHSAS, Droplet Measurement Technologies, Longmont, CO, USA) because the LAS 3340 was not available. Optical particle counters are designed to measure scattering, and the mathematical interpretation of the signal for highly absorbing biomass burning aerosols is described in detail in Sect. 3.

### 2.4.2 BBCES and CRDS flow system

As shown in Fig. 1b, the size-selected aerosol flow was diluted with 5.0 slpm of dry, particle-free zero air after exiting the DMA. The resulting 6.0 slpm of diluted sample flow was directed to the BBCES and CRDS channels (2.0 slpm per channel). Each cavity was constructed of aluminum tubing (22.1 mm ID for BBCES, 7.7 mm ID for CRDS), and all tubing between the DMA and the optical cavities was made of stainless steel or flexible silicone tubing to minimize electrostatic particle losses. To protect mirror cleanliness, particle-free zero air flowed over each cavity mirror at a rate of 55 – 80 sccm.

Two automated two-position valves (MDM-060DT-3; Hanbay Inc, Pointe-Claire, Quebec, Canada) allowed additional calibrations and zeros. The first two-position valve rotated to direct the sample flow through an aerosol filter, providing an aerosol-free reference measurement for the BBCES and CRDS cavities. The second two-position valve allowed pure gases to be introduced for periodic measurements. As shown in Fig. 1b, gas cylinders of He, $CO_2$, zero air (Norco, Inc., Missoula, MT, USA), and $NO_2$ in $N_2$ (Scott-Marrin, Inc., Riverside, CA, USA) were connected to automated valves and a mass flow controller. Pure $CO_2$ and He were used to characterize the mirror reflectivity by filling the cavities and purge volumes with each gas for 1.5 min at intervals of ~1 h. Pure zero air provided an aerosol-free extinction measurement, similar to the filter method described above. The cylinder containing ~3 ppmv $NO_2$ in $N_2$ provided additions of 0 – 72 ppbv $NO_2$ for flows of 0 – 100 sccm. Due to known discrepancies between the reported and actual concentrations of $NO_2$ calibration tanks (Chilton et al., 2005), this method was not used as a calibration, but rather an assessment of long-term stability of the instrument



with respect to the NO₂ spectral features. When the two-position valve was rotated for overflow of calibrant gas, the inlet sample flow and pressure were maintained using a mass flow controller and pump to minimize disturbances to the DMA and other instruments.

## 2.5 Shared aerosol inlet

The flow to the BBCES and CRDS channels was provided by a shared aerosol inlet, shown in Fig. 1c. The shared aerosol inlet consisted of ~30 m of 6.4 mm OD copper tubing connected to an inertial impactor with 50% cutpoint at an aerodynamic diameter of 1.0 um (TE296, Tisch Environmental, Cleves, OH), a silica gel dryer, an activated carbon denuder to remove $NO_y$ and $O_3$ (Washenfelder et al., 2015), and three available thermodenuders. The 2.0 volumetric L/min (vlpm) flow from the burn room was evenly split between one of the thermodenuders and a bypass channel. An automated two-position

valve (Hanbay Inc, Pointe-Claire, Quebec, Canada) alternately directed 1.0 vlpm of thermodenuded and 1.0 vlpm of fresh smoke to the BBCES aerosol inlet and to a mixing volume with dilution for a collection of other aerosol instruments (Manfred et al., 2018; Adler et al., 2019). The DMA, valves, MFCs, and all other components shown in Fig. 1 were controlled by custom LabVIEW software, and synchronized to the timing of the two-position valve shown in Fig. 1c as part of the NOAA shared aerosol inlet.

The inlet apparatus included three separate thermodenuders to enable comparison between different temperatures, but this paper reports data from only one. This denuder consisted of an 80 cm screen tube (1.3 cm ID) wrapped in activated charcoal fabric to remove volatilized organic components. The first 40 cm was heated to 250 °C and the remaining 40 cm served as a cool-down region. With a flow rate of 1.0 vlpm, the calculated residence time was 3.2 s and Reynolds number was 114. The denuders were similar to the design of Huffman et al. (2008), with the major difference that the heated section had

charcoal fabric to absorb the volatilized species. The throughput efficiency was found to be less than unity (86 ± 4% for particles between 100 and 300 nm) due to thermophoretic wall losses, but the analysis of the aerosol RI, which is an intrinsic property of the aerosol, is unaffected by these losses.

## 2.6 Black carbon and organic aerosol measurements

    Additional instruments sampled from the shared aerosol inlet's mixing volume, shown in Fig. 1c, and those data were

used to assess the fraction of black carbon in the total aerosol mass loading of smoke from each fire. A single particle soot photometer (SP2; Droplet Measurement Technologies, Longmont, CO, USA) measured the concentration of refractory black carbon aerosol, rBC (Schwarz et al., 2006; Schwarz et al., 2008). The rBC mass concentration was only retrieved in the mass range corresponding to particles with volume equivalent diameter of 90-550 nm, assuming a void-free density of 1.8g/cc. This range typically covers 90% or more of the accumulation-mode rBC emitted from wildfires. Refractory black carbon has been

shown to be experimentally consistent with elemental carbon measured after a thermal denuder within 15% (Kondo et al., 2011). The uncertainty of the black carbon mass at this concentration is 30%. A compact time-of-flight aerosol mass spectrometer (c-TOF AMS; Aerodyne Research Inc., Billerica, MA, USA) measured organic aerosol mass (Bahreini et al.,





2008; Bahreini et al., 2009). The uncertainty of the organic aerosol mass is typically ~38% (Bahreini et al., 2009), but was greater for biomass burning aerosol at the Fire Sciences Laboratory due to incomplete transmission of particles through the

AMS lens, beam spreading, and particle collection. Therefore, we use relative AMS masses in this analysis.

**2.7 BBCES instrument operation at the Fire Sciences Laboratory**

Each measurement day at the Fire Sciences Laboratory included two controlled burns, with 3 – 4 h of sampling of each burn. The instruments and inlet components were turned on at least 1 h prior to the first burn. Each sample cycle required 30 mins, and began with measurements of $CO_2$ and He to determine the BBCES mirror reflectivity, followed by measurement

of the dark noise of the CCD detector with the physical shutter closed. During these BBCES calibrations, the size distribution of the aerosol sample was measured by SMPS. Next, the BBCES and CRDS measured fresh and denuded (250 deg C) aerosol with mobility diameters of 150, 225, 300, 375, and 450 nm for two min each (20 min total). The aerosol mass loading was very high in the burn room (typically ~1g/m$^3$), but was reduced by a factor of ~100 by the DMA and the dilution flows prior to measurement by the BBCES. Therefore the two minute averaging time ensured sufficient precision in the spectra and the

particle counting, particularly at the larger diameter set points. These DMA diameter setpoints were chosen to span the observed aerosol size distribution of the smoke. Filtered sample air was measured before and after each set of five measurements. The five mobility diameter measurements allowed RI to be retrieved once every 10 min, alternating between fresh and denuded biomass burning smoke. Between the two daily fires, $NO_2$ additions were made to assess the spectrometer stability. The spectrometer wavelength and resolution was calibrated with the output of a Hg lamp before the first burn, between

the first and second burns, and after the second burn.

After the FIREX 2016 study, the BBCES extinction measurements and complex RI retrievals were validated using standard samples of polystyrene latex spheres (PSL; Nanosphere size standards, Thermo Fisher Scientific Inc., Waltham, MA, USA) and ammonium sulfate (Sigma Aldrich, St. Louis, MO, USA). For these measurements, the aerosols were generated using a custom-built atomizer, and dilution make-up flow was provided by scrubbed and dried air from a compressor. Five or

six aerosol sizes were selected, spanning a similar range to those sampled during FIREX. The sampling scheme shown in Figs. 1a and 1b remained the same, but the shared inlet shown in Fig. 1c was not used. Particle-free air and mirror reflectivity measurements were performed before and after each set of aerosol sizes.

**3 Data Analysis**

**3.1 Determination of aerosol optical extinction and mirror reflectivity**

The aerosol optical extinction, $\alpha(\lambda)$, can be determined from the observed change in light intensity in the cavity according to

$$\alpha(\lambda) = d_L \left( \frac{1-R(\lambda)}{d} + \alpha_{Rayleigh,ZA}(\lambda) \right) \left( \frac{I_{ZA}(\lambda) - I(\lambda)}{I(\lambda)} \right) \tag{1}$$





where $\lambda$ is wavelength, $d_L$ is the ratio of the physical cell length to the sample cell length, $R(\lambda)$ is the measured mirror reflectivity, $d$ is the physical cell length, $\alpha_{Rayleigh,ZA}(\lambda)$ is the Rayleigh scattering of zero air, $I_{ZA}(\lambda)$ is the reference intensity spectrum, and $I(\lambda)$ is the measured intensity spectrum (Washenfelder et al., 2013). The extinction cross section, $\sigma(\lambda)$, is defined

as the optical extinction divided by the total number concentration of aerosol particles, N, for the given size distribution:

$$\sigma(\lambda) = \frac{\alpha(\lambda)}{N} \qquad (2)$$

The most common method for determining $R(\lambda)$ in Eq. (1) is through extinction measurements of two gases with known Rayleigh scattering cross sections, typically He and $N_2$ or zero air (Washenfelder et al., 2008). Previous BBCES applications in the UV wavelength region often used He and $N_2$, which have substantially different cross sections

(Washenfelder et al., 2013; Flores et al., 2014a; Bluvshtein et al., 2017; Zhao et al., 2017). However, Rayleigh scattering is proportional to $\lambda^{-4}$ and more accurate characterizations of mirror reflectivity at visible wavelengths can be achieved by using a gas with a larger Rayleigh scattering cross section than $N_2$ or $O_2$, particularly when the mirror reflectivity is not very high. We selected $CO_2$ as a stable, inexpensive, non-toxic gas with a large Rayleigh scattering cross section. The Rayleigh scattering cross section of $CO_2$ has been reported (Shardanand and Rao, 1977; Sneep and Ubachs, 2005; He et al., 2018), but it is not as

well-known as $N_2$ or $O_2$. We determined it here by introducing mixtures of $CO_2$ in He to the BBCES instrument, measuring the extinction spectrum for each addition, and deriving the slope of the linear fit of the extinction relative to the $CO_2$ number density.

### 3.2 Correction of optical extinction for optical intensity and spectral drift

Spectral fitting of trace gas absorbers in BBCES extinction spectra often includes a 4th-order polynomial to account

for drifts in the cavity, spectrometer, and light source intensity (Platt et al., 2009; Thalman and Volkamer, 2010; Min et al., 2016). While this can be effective for trace gases with highly structured absorbing features measured using mirrors with smoothly varying reflectivity, it is not appropriate for smoothly-varying aerosol extinction spectra or for cavity mirrors with reflectivity that varies strongly with wavelength (see Fig. S1). Here, we describe a new method to explicitly account for drifts in the cavity, spectrometer, and light source intensity that affected the calculated $\alpha(\lambda)$ extinction for the visible channel.

We determined that each measured spectrum may be affected by drift in the light source intensity, drift in the spectrometer dark current background counts, and a wavelength shift incurred by the temperature-dependent drifts in the spectrometer optics, and that these can each be represented by a scalar parameter. Using a nonlinear Levenberg-Marquardt least-squares fitting algorithm, the scalar parameters for these different types of drift were fitted and the drifts removed. The fitting algorithm was tested successfully on spectra of zero air and found to remove nearly all the structure due to intensity and

spectrometer drift. This correction was subsequently used on all extinction spectra. Further details for this correction can be found in the Supplemental Information and in Fig. S2.





### 3.3 Complex refractive index retrieval

The aerosol optical extinction derived in Eq. (1) depends on wavelength, aerosol size distribution, and the complex refractive index. Measurements of aerosol optical extinction for different size-selections can be used to retrieve $n(\lambda)$ and $k(\lambda)$

(Washenfelder et al., 2013; Flores et al., 2014a; He et al., 2018) with assumptions that the aerosol population is internally and externally well-mixed and that the refractive index does not vary systematically with size. For PSL and ammonium sulfate aerosol, this approach was used as described previously (Washenfelder et al., 2013; Flores et al., 2014a). Each two-minute group of spectra for a single aerosol population was averaged to determine $I(\lambda)$. The aerosol extinction cross section, $\sigma(\lambda)$, was then calculated using Eqs. 1 and 2, with values of $I_{ZA}(\lambda)$ and $R(\lambda)$ linearly interpolated from the nearest measurements. The set

of aerosol extinction cross section spectra ($\sigma_{Dp=150\ nm}(\lambda)$, $\sigma_{Dp=225\ nm}(\lambda)$, …) and the average aerosol size distribution $N(D_p)$ for each time interval can then be used to retrieve the complex refractive index, $n(\lambda) + ik(\lambda)$ using a scattering theory.

We use a retrieval algorithm that calculates the expected extinction cross section for a given wavelength, a given RI, and the measured $N(D_p)$ and compares it to the measured extinction cross section. Since the RI consists of a scattering and an absorbing component, $n$ and $k$, at least two extinction measurements are required to retrieve these two variables (Bluvshtein et

al., 2012). We use five diameter setpoints to increase the accuracy of the retrieval. A least-squares minimization fit is used to identify the $n$ and $k$ values that minimize $X^2$ for each wavelength

$$\chi^2(\lambda) = \sum_{Dp}^{N=5} \left( \frac{\sigma_{calculated}(n,k,\lambda,Dp) - \sigma_{measured}(\lambda,Dp)}{\sigma_{measured}(\lambda,Dp)} \right)^2 \tag{3}$$

where $\sigma$ represents the extinction cross section from Eq. (2).

Previous work has assumed spherical aerosol populations that are well represented by Mie theory and have easily

measured size distributions (Washenfelder et al., 2013; Flores et al., 2014a; He et al., 2018). The biomass burning aerosol produced from many of the fires at the Fire Sciences Laboratory were fractal and strongly absorbing. This required two improvements to the retrieval algorithms, described further below: 1) incorporation of two methods for size distribution measurements (the OPC and the SMPS) to better characterize $N(D_p)$; and 2) incorporation of two scattering theories, Mie and Rayleigh-Debye-Gans, to represent both spherical and fractal aerosol.

### 3.3.1 Treatment of the aerosol size distribution

The DMA transmits ionized particles of a specified mobility to charge ratio, resulting in a small number of doubly- and triply-charged particles that appear as particles with mobility diameters approximately 2× and 3× greater than expected. The relative contribution of the singly-, doubly-, and triply-charged particles to the total size distribution depends on the ambient aerosol size distribution, as well as the ratio of sheath flow to aerosol flow in the DMA. To determine the aerosol size

distribution, we iteratively adjusted an assumed input size distribution transmitted through the DMA transfer function, until the theoretical particle concentration produced by the inversion matched that measured by the CPC (Twomey, 1975; Markowski, 1987). Once the input size distribution was determined, the DMA transfer function was used to estimate the size distribution of the aerosol existing the DMA as the voltage was changed to select different peak diameters.





For spherical, non-absorbing particles, such as ammonium sulfate aerosol, we expect the SMPS scans and inversions
to yield accurate size distributions, as the contribution of multiply-charged particles is straightforward to calculate for spherical
particles with well-known mobility diameters. If the aerosol is significantly non-spherical, such as fractal black carbon
expected from some biomass burning smoke, then the transmission through the DMA is affected by the aerodynamic resistance
of the sheath flow, the contribution of multiply-charged particles is difficult to ascertain, and the resulting size-selected aerosol
may not match the actual particle size. We therefore do not attempt any correction to the SMPS scans for black carbon aerosol;
the inverted size distributions represent electrical mobility diameter, rather than geometric diameter.

An alternate aerosol sizing instrument is the OPC, which sizes particles by measuring the light scattered into a wide-
angle lens or mirror mounted perpendicularly to the direction of laser propagation. The integrated side-scattering intensity
increases monotonically with particle size for non-absorbing particles. However, a theoretical correction must be made for
slightly- or highly-absorbing particles. By calculating the scattered light expected as a function of angle, using either Mie or
RDG theory (see Sect. 3.3.2), one can integrate over the solid angle of the LAS measurement optics to get a corrected LAS
size distribution. We discuss this correction further in Sect. 4.6.

Particle losses between the DMA and the BBCES instrument were calculated theoretically (von der Weiden et al.,
2009) as a function of particle diameter, and were accounted for in the size distribution calculations. These losses were
generally small, with fewer than 1% loss in the relevant particle diameter range of 100 – 800 nm for this analysis.

**3.3.2 Mie theory and Rayleigh-Debye-Gans theory**

Two scattering theories are used in this paper to characterize the optical properties of the measured aerosol. Mie
theory is a solution to Maxwell's equations that describes the interaction of light with homogeneous, spherical particles, when
the diameter of the sphere is similar to the wavelength of light (Bohren and Huffman, 1983). The theory is valid when the
dimensionless size parameter ($x = \pi d / \lambda$) is approximately 1. It is a series approximation that allows a mathematical
representation of light with spheres, concentric spheres, and clusters of spheres. We adapted the Fortran code presented in
Bohren and Huffman (1983) for use in Igor Pro (Igor Pro; WaveMetrics, Inc., Lake Oswego, OR, USA). The code calculates
light scattering and absorption for homogeneous spheres or homogeneous spheres with a concentric coating.

Rayleigh-Debye-Gans theory has often been applied to calculate the interaction of light with fractal aggregates from
biomass burning (Sorensen, 2001). Spherical approximations and Mie theory are poor representations of these particles, and
more detailed models such as the Discrete Dipole Approximation are very computationally expensive. Rayleigh-Debye-Gans
theory divides a particle into small volume elements, which are each treated as independent Rayleigh scatterers. Fractal
aggregates can be represented by the relatively simple parameterization

$$N = k_0 \left(\frac{R_g}{a}\right)^{D_f} \tag{4}$$

where $N$ is the number of monomer spherules per aggregate, $k_0$ is the fractal prefactor, $R_g$ is the radius of gyration, $a$ is the
monomer radius, and $D_f$ is the fractal dimension (Sorensen, 2001; Smith and Grainger, 2014). The theory is valid when the





relative refractive index and the dimensionless size parameter of the monomer can be described as $|n - 1| < 1$ and $x \leq 0.3$ (Farias et al., 1996). Code to calculate the expected extinction cross section as a function of the input parameters was written in Igor Pro, following Sorensen (2001).

**3.4 Precision and accuracy of measured aerosol extinction and retrieved RI**

The accuracy of the measured aerosol extinction, $(\lambda)$, was determined by propagating the known uncertainties of the measurement. The precision in the $I(\lambda)$ spectrum is shot noise limited at the CCD. Figure S1b demonstrates that error from this photon-counting is negligible compared to the other uncertainties. The majority of the error is attributed to uncertainty in the Rayleigh scattering cross sections of zero air ($\pm 2\%$) and $CO_2$ ($\pm 2\%$), as well as uncertainty in the measured temperature ($\pm 0.5\%$) and pressure ($\pm 0.5\%$), and flow rates ($\pm 0.3\%$) (Washenfelder et al., 2013). Summed in quadrature, the total calculated

uncertainty in $\alpha(\lambda)$ is approximately $\pm 3\%$, with some wavelength dependence. The uncertainty in the extinction cross section, $\sigma(\lambda)$ as shown in Eq. (2), is derived from summing in quadrature the error in $\alpha(\lambda)$ and the error in $N$. The error in the particle counting, $N$, by the CPC is due to the uncertainty in the flow rate and the variability in the number of particles entering the CPC. The last factor is highest for the largest diameter set points due to the low number of particles, and thus the uncertainty in $N$ typically lay between $\pm 5$ and $\pm 10\%$ and was highest at the largest aerosol diameter set points. The standard deviations for

all measurements are indicated in the figures by shaded error bars. The derived lower detection limit as a function of wavelength is shown in Fig. S1b, and spans $1 \times 10^{-8} – 7 \times 10^{-8}$ cm$^{-1}$. Extinction values measured during FIREX 2016 were generally greater than $1 \times 10^{-7}$ cm$^{-1}$, and therefore within the measureable range of the BBCES.

        To ascertain the uncertainty in the retrieved RI, we retrieve values three times: once for the best estimate of $\sigma(\lambda)$, and once for the upper and lower bounds of $\sigma(\lambda)$. This is likely a conservative estimate of the total uncertainty, and is shown in

shaded error bars in the figures. The RI was retrieved with a resolution of 0.01 in both $n$ and $k$, so if the upper or lower bound of the retrieved RI was within 0.01 of the best estimate, those error bars appear to be zero. In that case, the error should be interpreted as being no more than 0.01.

**4 Results and Discussion**

        We present the results of measurements taken during and after the FIREX 2016 deployment. First, the measurement

of the $CO_2$ Rayleigh scattering cross section between 360 and 720 nm is presented, along with the measurement of daily additions of known concentrations of $NO_2$ throughout the duration of the Fire Lab experiments, to ascertain the long term stability and accuracy of the BBCES extinction cross section measurements. Next, we present retrievals of the RI of known laboratory standards, PSL and ammonium sulfate, across a wider wavelength range than has previously been measured. We use these measurements to assess the reliability of the two size distribution methods. We examine measurements of two

contrasting fires from the Fire Lab, one dominated by spherical brown carbon particles, the other by fractal black carbon particles, and demonstrate that the first can be reasonably characterized using Mie theory, while the second cannot, and requires




Rayleigh-Debye-Gans theory. These two fires are the same presented in Manfred et al. (2018), and we follow them in denoting the two fires Fire A and Fire B, respectively. Finally, RIs were retrieved for 13 fires assuming Mie theory, and the quality of the fits is related to the black carbon content of the smoke, demonstrating that smoke with higher BC generally cannot be fit
well with Mie theory.

### 4.1 $CO_2$ Rayleigh scattering cross section

Figure 2 shows the measured $CO_2$ Rayleigh scattering extinction spectra, determined from six fractional concentrations of $CO_2$ in He that evenly span $0 - 1$. The mirror reflectivity, $R(\lambda)$, was determined here using pure He and zero air standards. The $CO_2$ extinction spectrum, $\alpha(\lambda)$, at each concentration was determined using Eq. (1), with He replacing zero
air as the reference gas. Spectral peaks attributed to absorption by $O_2$ near 688 nm (Greenblatt et al., 1990) and the $O_4$ oxygen dimer near 477, 577, and 630 nm (Hermans, 2011) were fitted and removed prior to further analysis. At each wavelength, the measured extinction (units of $cm^{-1}$) was plotted against the number density of $CO_2$ (units of $cm^{-3}$), and the slope of the linear fit is the Rayleigh scattering cross section (units of $cm^2$).

The $CO_2$ Rayleigh scattering cross section determined for the two BBCES channels and CRDS channel agree well,
as shown in Fig. 2. A power law fit to the BBCES data from $360 - 720$ nm is parameterized as $\sigma_{Rayleigh,CO2} = 1.43 \times 10^{-15} \times \lambda^{-4.036}$. Three literature values for the $CO_2$ Rayleigh scattering cross section are also plotted in Fig. 2. These were experimentally determined using a scattering chamber (Shardanand and Rao, 1977), CRDS (Sneep and Ubachs, 2005), and BBCES (He et al., 2018). Our power law fit is consistent with all prior measurements within 10% at all wavelengths, and is within 5% of the two more recent studies at lower wavelengths where the Rayleigh scattering is larger. We use the power law parameterization of
$CO_2$ Rayleigh scattering cross section for the retrievals described below.

### 4.2 Standard additions of $NO_2$

To assess the long-term stability of the instrument, we introduced concentrations of $NO_2$ during each of the 16 measurement days at the Fire Sciences Laboratory, using the flow system shown in Fig. 1b. Each set of additions included four $NO_2$ concentrations from 0 to 72 ppbv. Measured $NO_2$ number densities were retrieved by fitting the $NO_2$ absorption cross
section (Vandaele et al., 1998) to the extinction, $\alpha(\lambda)$, with shift and stretch of the literature spectrum to account for discrepancies in the BBCES wavelength calibration (Min et al., 2016). A small amount of HONO was observed in the UV channel, but has distinct spectral peaks, and did not interfere with the $NO_2$ fits. The measured $NO_2$ concentrations are compared to the nominal concentrations determined by the flow rates, and shown in Fig. S3. The $r^2$ values are greater than 0.99 and the slopes vary from $0.8 - 1$, with a slight downward drift in both parameters over time. This could be due to conversion of $NO_2$,
as the BBCES is designed to minimize particle losses, not $NO_2$ conversion on surfaces. It is also possible that the $NO_2$ calibration tank concentration changes over time, as the output of these tanks is known to deviate significantly from the manufacturer's specified concentration (Chilton et al., 2005). The two fires examined in this paper, Fires A and B, occurred on 31 Oct and 1 Nov 2016, and Fig. S3 shows agreement of $\pm7\%$ with the nominal $NO_2$ concentration on those two days.



### 4.3 Refractive index retrieval for PSL

We atomized five aqueous solutions of monodisperse PSL with diameters from 150 to 400 nm, and passed the resulting aerosol flow through a DMA to remove any clumped particles or small spheres of surfactant (Miles et al., 2010; Thalman and Volkamer, 2010). The size-selected aerosol was monitored by a UHSAS OPC and found to be consistent within uncertainty with the manufacturer-specified mode diameter and Gaussian distribution. SMPS-derived size distributions of PSL were not measured, due to their narrow diameter distribution. The intensity spectra and size distributions were averaged for

approximately 90 s at each diameter setpoint, and extinction cross sections were derived using Eqs. (1) and (2), as described in Sects. 4.1 and 4.2. To minimize errors due to changes in the light source and optical stability, we measured the mirror reflectivity before and after each PSL diameter set point, in addition to using the fitting algorithm described in Sect. 3.2 and S1.

          The resulting extinction cross sections for the five PSL diameters are shown in Fig. 3a. Calculated extinction cross

sections from the measured OPC size distribution and the RI reported by Nikolov and Ivanov (2000) are also shown, and these agree with our measured extinction cross sections within the instrument uncertainty. The largest error bars appear in the $D_p$ = 400 nm extinction cross section. It was difficult to produce PSL solutions with sufficiently high concentrations at the larger diameters, and the extinction cross section, which is the total extinction divided by the total particle counts, is therefore highly sensitive to the accuracy of the particle counting. To reduce error in the RI retrieval, we exclude the 400 nm spectrum from

the RI retrieval.

          The retrieved RI is shown in Fig. 3b. We find that the scattering component varies from $n$ = 1.65 to 1.57 between 360 and 700 nm, while the absorbing component is constant at $k$ = 0 throughout, as expected for purely scattering particles. We empirically fit $n$ to a 3$^{rd}$ order polynomial function of wavelength as $n = 2.30 - 0.0035 \times \lambda + (6.05 \times 10^{-6}) \times \lambda^2 - (3.62 \times 10^{-6}) \times \lambda^3$ across the measured spectral range. Nikolov and Ivanov (2000) measured the RI of PSL spheres using a direct scattering

photodetector and found a gradual decrease from $1.62 + 0i$ to $1.59 + 0i$ over the wavelength region 450 to 700 nm, as shown in Fig. 3b. Several other recent experimental studies (Garvey and Pinnick, 1983; Abo Riziq et al., 2007; Dinar et al., 2008; Washenfelder et al., 2013) at 390 and 532 nm are also shown and agree well with our results.

### 4.4 Refractive index retrieval for ammonium sulfate

          Figure 4 shows the retrieved RI for ammonium sulfate aerosol, using the two different size distribution methods.

Ammonium sulfate particles were atomized from a 6.0 g/L aqueous solution of ammonium sulfate in distilled water, and then dried to less than 5% RH to minimize the water content. Extinction cross sections were determined at diameter set points from 150 – 450 nm in 75 nm increments. The aerosol size distribution was measured continuously after the DMA by a UHSAS OPC, which was independently calibrated for ammonium sulfate particles prior to the experiment (Kupc et al., 2018). Additionally, SMPS scans were taken before and after each set of aerosol diameter setpoints, and an inversion algorithm was

used to derive the full size distribution of the atomized aerosol entering the DMA. Spurious peaks in the size distribution at

 

diameters greater than 750 nm were removed. Section S2 describes this correction further. The DMA transfer function, was then used to determine the aerosol size distribution at each diameter set point.

The RI values in Fig. 4 range from 1.55 + 0*i* at 360 nm to 1.40 + 0.01*i* at 720 nm, though the error bars for the imaginary component span 0*i*, and thus cannot be distinguished from a non-absorbing aerosol. The results from the two size
distribution methods agree well, indicating that the SMPS size distribution is an accurate assessment of the aerosol population. We empirically fit the *n* derived by OPC as a 3$^{rd}$ order polynomial function of wavelength as $n = 0.69 + 0.0052 \times \lambda - (9.85 \times 10^{-6}) \times \lambda^2 + (5.51 \times 10^{-6}) \times \lambda^3$ across the entire spectral range. Recent literature values measured by CRDS at 355, 390, 405, and 532 nm are shown and summarized in Washenfelder et al. (2013). Few measurements of ammonium sulfate RI have been reported at wavelengths greater than 532 nm. The calculated uncertainty is higher at longer wavelengths due to increased
uncertainty in the Rayleigh scattering at those wavelengths.

**4.5 Refractive index retrieval for aerosol dominated by brown carbon (Fire A)**

Fire A (#086) was a lodgepole pine (*Pinus contorta*) fire, with a mixture of small logs, litter duff, and canopy branches with low moisture content, intended to be representative of wildfire fuel composition. The smoke was dominated by brown carbon with a small contribution from fractal soot-like particles, as shown by the large decrease in number and mode volume
when the aerosol was thermodenuded (Manfred et al., 2018). This implies that the majority of the particles consisted of volatile organic matter that was efficiently vaporized in the thermodenuder. We assume that the volatile organic matter has spherical or near-spherical morphology that can be represented using Mie theory.

The retrieved RI values for undenuded aerosol measured 1 h after fire ignition are shown in Fig. 5a for one 10-min measurement cycle. Both SMPS and OPC size distributions were available for this set of extinction cross sections, and two
diameters are displayed in Fig. 5b. However, there are some differences between the two size distribution measurements. First, as in the ammonium sulfate retrieval in Sect. 4.4, the SMPS inverted full size distribution has some spurious values at diameters above 550 nm which are not observed in the OPC, and which skew the size distributions to higher diameters. Therefore, we again apply an upper diameter threshold, above which the SMPS size distributions are set to zero. Second, the OPC size distribution for every diameter set point shows a small peak near 100 nm. That these small particles passed through the DMA
at larger diameter set points suggests that they are small fractal particles, which were selected by the DMA due to their relatively large aerodynamic diameter. Scanning electron microscopy images from the denuded channel for this fire confirm the presence of small fractal particles (Manfred et al., 2018). These particles are quite small and therefore do not contribute much to the total aerosol extinction, but for the larger diameter set points they comprise a significant percentage of the total number of particles. Therefore, we must explicitly include these small particles as part of the size distribution.
Finally, it can be seen in Fig. 5b that the main mode diameter for each set point is approximately 5% higher in the OPC than the intended set point, as indicated by the mode diameter in the SMPS. The LAS OPC is calibrated for ammonium sulfate particles, assuming an RI of 1.52 + 0*i*. If the RI of these brown carbon particles has a higher scattering component in the RI than 1.52, then the OPC will interpret this increased light scattering as a larger particle. This also holds true for the small





~100 nm peaks, which likely consists of BC particles with a RI very different from that of ammonium sulfate. Therefore, the
sizing of these particles by the OPC may be somewhat incorrect.

Despite these caveats, we use the LAS size distribution without further modification for the RI retrieval in Fig. 5a.
We find that the real component of the retrieved RI varies from 1.55 to 1.60, while the imaginary component steadily increases
from ~0$i$ to 0.25$i$ as the wavelength decreases from 550 – 360 nm. This is highly characteristic of brown carbon aerosol and
consistent with previous estimates of the complex refractive index of fresh biomass burning aerosol summarized in Bluvshtein
et al. (2017). Furthermore, these observations differ significantly from the measured RI of ~1.8 + 0.01$i$ for spherical black
carbon described as tar balls by Chakrabarty et al. (2010), indicating that the aerosol population measured here does not have
a significant contribution from tar balls. At the OPC laser wavelength of 663 nm, the retrieved RI is (1.61 ± 0.14) + (0.00 ±
0.06)$i$. Particles with this RI will scatter light slightly more efficiently than ammonium sulfate particles, and therefore the LAS
will attribute this scattered light to a slightly larger particle. Therefore, this result is consistent with our finding that the LAS
mode diameters are slightly higher than the diameter set point. However, we do not attempt to correct the LAS diameters
further, as that analysis is complicated by the small number of fractal particles also observed. Manfred et al. (2018) assumed
a typical RI for brown carbon (Dinar et al., 2008), shown in filled squares at 375 and 405 nm, to model the measured phase
function from this fire, with good results. We find good agreement with those assumed RIs within instrumental uncertainty.

## 4.6 Refractive index retrieval for aerosol dominated by black carbon (Fire B)

Fire B (#085) was a sage (*Artemisia*) fire that produced smoke with a high black carbon content, as evidenced by the
minimal change in particle counts and volume between the fresh and thermodenuded smoke (Manfred et al., 2018).
Furthermore, Manfred et al. (2018) demonstrated that the measured phase function for this aerosol population could not be
accurately fit assuming Mie theory and a typical brown carbon RI. Instead, they used Rayleigh-Debye-Gans theory, along with
two different parameterizations of the fractal particles to fit the measurement. As described in Sect. 3.3.2, RDG theory treats
fractal particles as an aggregate of many smaller monomers, and the input parameters describe the number of monomers, their
configuration in the aggregate, and the extinction properties of each monomer.

Unfortunately, neither size distribution method used here can accurately constrain these parameters well. The SMPS
size distributions are inaccurate because fractal particles have mobility diameters that differ significantly from their geometric
diameters. The OPC size distributions yield more useful information because they measure scattered light within a certain solid
angle range, which is later attributed to a certain particle diameter. Only purely scattering and very slightly absorbing particles
display monotonically increasing scattering as a function of increasing particle diameter. Highly or moderately-absorbing
particles tend to side-scatter significantly less than scattering particles to the angles the LAS is most sensitive to, and the
scattering is often non-monotonically increasing as particle size increases (Szymanski et al., 2009). Therefore, a large diameter
range of absorbing particles have similar scattering properties and cannot be distinguished, as illustrated in Fig. S5. However,
we can make an adjustment to the measured OPC size distributions. As described in Sect. 3.3.2, we use RDG theory to predict
the side-scatter into the OPC collection optics for particles as a function of the number of monomers, $N_p$, while holding the





other parameters in Eq. (4) constant. This allows us to treat the nominal OPC size distribution as a distribution of monomers in the fractal particle. We can then use the same RDG parameters to estimate the extinction cross section and compare it to the measured spectra. Because the RI retrieval is so sensitive to an accurate representation of the particle size, we do not attempt
to fit any of the RDG parameters. Instead, we show the expected cross sections for two plausible RDG parameterizations of BC particles.

Figure 6 shows the measured extinction cross sections for the five diameter set points. These spectra were taken 1 h after the fire was ignited, and represent the aerosol that passed through the thermodender. Therefore, the majority of organic coatings that might complicate the analysis are removed prior to measurement. Following Manfred et al. (2018), we use two
parameterizations. Both assume an RI for black carbon of $1.95 + 0.8i$ (Bond and Bergstrom, 2006), and an individual monomer diameter of 50 nm. The first is meant to represent fractal particles from fossil fuel combustion. It assumes a prefactor of 1.2 and a fractal dimension of 1.75 (Sorensen, 2001). The second represents sage biomass burning, and consists of a prefactor of 2.56 and a fractal dimension of 1.79 (Chakrabarty et al., 2006). The two parameterizations are shown in Figs. 6a and 6b. The fossil fuel parameterization yields the closer approximation of the measured data, but the measurement lies in between the two
parameterizations. This is consistent with the results of Manfred et al. (2018), in which the measured phase function for thermodenuded smoke from Fire B lay in between the two parameterizations.

**4.7 Summary of refractive index retrievals for 13 fires during FIREX**

We place the Fire A and B retrievals in the context of the other FIREX measurements by analyzing the quality of the RI retrieval method assuming Mie theory, and relating these values to the fraction of aerosol that is black carbon. The BC /
(BC + OA) fraction was calculated relative to the largest BC fraction for 13 fires for which the data was available from the average SP2 black carbon mass and the relative AMS organic aerosol mass, and the error bars were derived from propagating the two instrument uncertainties. Only aerosol that bypassed the thermodenuder was used in this analysis, in order to quantify fresh non-denuded smoke. We elected to evaluate the RI retrieval as a function of the BC / (BC + OA) ratio instead of the MCE, as BC / (BC + OA) has been shown to more strongly correlate with single scattering albedo and the Ångström absorption
coefficient (Pokhrel et al., 2016). The RI retrievals were performed using the SMPS size distributions, as the LAS data was often not available. Therefore, this analysis is also a function of how well the SMPS size distributions characterize the aerosol geometric diameter. We report the $\chi^2$ of the RI retrieval at 475 nm, where the instrument had high mirror reflectivity and therefore good precision. The reported $\chi^2$ for each fire is from a single set of measurements through the bypass channel, approximately 1 h after the fire start, and therefore represents well-mixed smoke. The derived real part of the RI for the low-
BC fires ranged from 1.54 to 1.69 at 475 nm with an average value of $1.635 \pm 0.056$, and the imaginary part ranged from $0i$ to $0.23i$ with an average value of $0.06i \pm 0.12$.

Figure 7 shows a correlation between the relative fraction of BC in the total aerosol mass and the quality of the RI retrieval assuming Mie theory and spherical particles well-characterized by the SMPS. Fire B has the highest relative BC fraction, defined here as 1, and is poorly described by Mie theory, with a $\chi^2$ of 1.3. This finding is consistent with Smith and





Grainger (2014), who found that there is no refractive index that suitably describes fractal BC if those particles are treated as spheres with Mie theory.  Fire A has a lower relative BC fraction and is better described by Mie theory, consistent with Sects. 4.5 and 4.6. Most of the other eleven fires analyzed here have a smaller BC fraction than Fire A and a correspondingly lower $\chi^2$ value. This indicates that the majority of smoke sampled during FIREX had relatively low BC and could reasonably be analyzed using Mie theory. However, we note that this freshly-emitted smoke was from small scale burns under controlled

conditions, and may not represent the full range of smoke aerosol types in ambient aerosol, particularly further downwind of the fire where coagulation and fractal aggregate collapse can alter the particle morphology. Furthermore, the poor fit quality of the high BC fire (Fire B) shows that Mie theory is not appropriate for all smoke, especially smoke with a high BC fraction. Since BC fraction is not a metric that is easily quantified with remote sensing, further work is still needed to link this parameterization to remote sensing measurements to predict when Mie theory representation of wildfire smoke is appropriate.

**5 Summary and conclusions**

This paper describes the development of a new broadband cavity enhanced spectrometer that measures aerosol extinction over a very broadband wavelength region in the ultraviolet and visible spectral regions. It was outfitted with an automated flow system for size-selected retrievals of the complex refractive index of biomass burning aerosol. After measuring the $CO_2$ Rayleigh scattering cross section across the 360 – 720 nm spectral region, and confirming the long-term stability and

accuracy of the instrument with repeated additions of $NO_2$, we demonstrated the effectiveness of the RI retrieval algorithm on two laboratory standard aerosols: PSL and ammonium sulfate. Good agreement was found for each standard with other literature values. We demonstrate that the OPC is an effective method for quantifying the size distribution of the generated aerosol, but that it is quite sensitive itself to aerosol refractive index. We also demonstrate that with a simple diameter threshold cutoff to remove artifacts, the SMPS can provide accurate size distributions for spherical particles which yield retrievals

consistent with literature values.

This instrument was deployed to the Missoula Fire Sciences Laboratory for the 2016 FIREX campaign, where it measured real biomass burning aerosol from a series of controlled burns of North American fuels. We presented measured extinction cross sections from two representative fires: lodgepole pine, which produced smoke dominated by brown carbon, and sage, which produced smoke dominated by black carbon. The optical properties of the BrC-dominated smoke could be

modeled with Mie theory and a wavelength-dependent RI that increased its absorption component as wavelength decreases. This behavior is consistent with other observations of BrC, and demonstrates that this BBCES instrument and retrieval algorithm is capable of measuring biomass burning aerosol.

The BC-dominated smoke, on the other hand, could not be modeled using Mie theory, and was better represented by Rayleigh-Debye-Gans theory. Two possible RDG parameterizations were assessed to quantify their ability to reproduce the

observations. We found that the measurement lay somewhere between the two parameterizations. Due to the large number of unknown variables, and the absence of a reliable metric of the particle morphology, we do not attempt to further characterize



the optical properties of this BC smoke. However, we find that a reasonable parameterization of RDG theory can largely reproduce the measured results, and recommend this parameterization for further characterization of BC-dominated fires.

We retrieved RIs for eleven additional fires and found a correlation between the quality of the RI retrieval fit and the fraction of BC, with high BC fires, such as Fire B, poorly described by Mie theory, and those with lower BC content to be better described by Mie theory. We found that the majority of smoke measurements had fairly low BC content, and the RIs could therefore be reasonably retrieved. However, remote sensing retrievals of biomass burning aerosol must account for the optical properties of smoke with varying amounts of BC, and Mie theory is therefore not always an appropriate theory to use.

**Data availability**

All data is available upon request to the corresponding author (caroline.womack@noaa.gov).

**Author contribution**

CCW, KMM, NLW, GA, AF, KDL, AMM, JPS, and RAW collected and analyzed the Fire Lab data. CAB advised the data collection of the laboratory standards and provided instrumentation. CCW, SSB, and RAW conceptualized the experiment, data collection, and data analysis. CCW wrote the data analysis code. KMM advised the data analysis. CCW and RAW wrote 555 the manuscript with contributions from all authors.

**Competing interests**

The authors declare that they have no conflict of interest.

**Acknowledgements**

We thank Yinon Rudich and Quanfu He for high finesse cavity mirrors and technical discussions about their properties. We 560 thank Agnieszka Kupc and Karl Froyd for the loan of the LAS and providing calibration information. We thank Jim Roberts, Carsten Warneke, and Bob Yokelson for coordinating the Fire Lab campaign, and the Fire Sciences Laboratory in Missoula for hosting the campaign. This project was supported by the NOAA Atmospheric Chemistry, Carbon, and Climate Program (AC4).



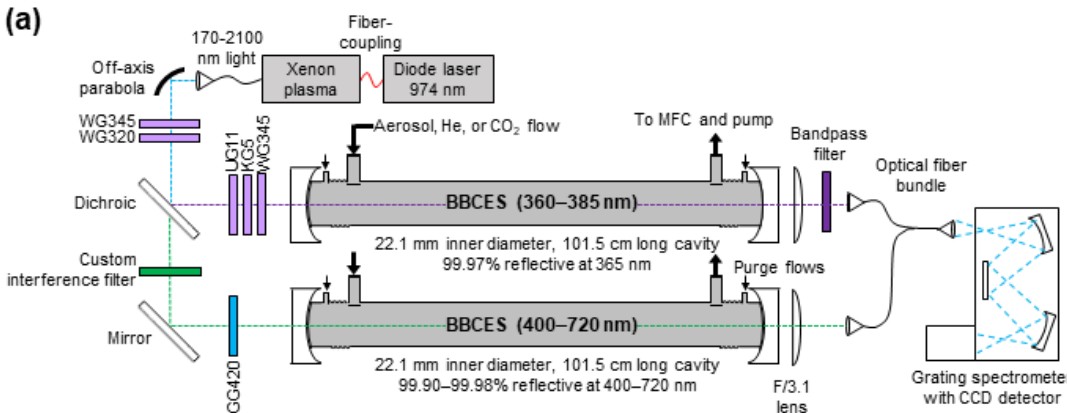

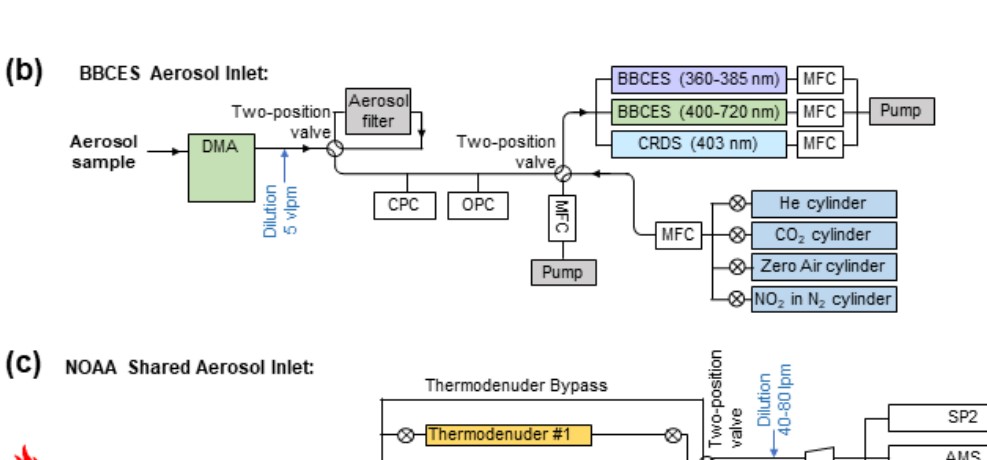


**Figure 1. A schematic of the BBCES instrument and inlet at the Fire Lab. (a) The optical components of the BBCES instrument include a laser-driven arc lamp, off-axis parabola, colored glass filters, dichroic beamsplitter, two BBCES cavities, and grating spectrometer. (b) The automated flow system developed for the Fire Lab. Aerosol is size-selected by a DMA, with the number concentration continuously measured by a CPC and the size distribution periodically measured by an OPC. The flow is evenly**


**divided between the two BBCES channels and the CRDS channel. An automated two-position valve allows flow to be directed through a filter to measure particle-free air, and a second two-position valve allows the introduction of calibration gases. (c) The common inlet shared by multiple aerosol instruments at the Fire Lab, including the SP2 and the AMS. Smoke was sampled at 2 vlpm through an impactor (1 um cut-point), silica gel dryer, and NOy scrubber, before traveling through a bypass channel or thermodenuder. A two-way valve alternately directed fresh and denuded aerosol to the BBCES/CRDS at 10 min intervals.**





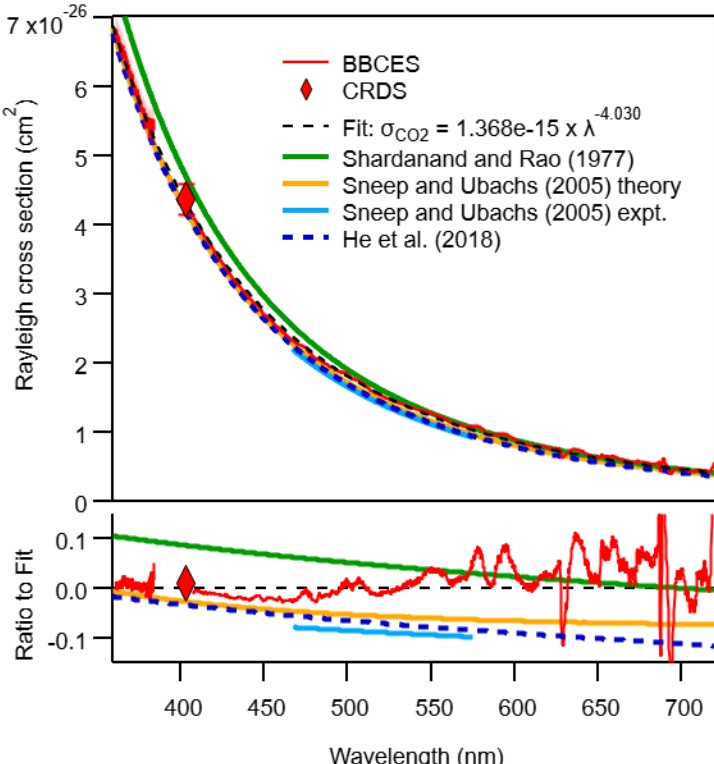


**Figure 2. The measured Rayleigh scattering cross section of CO₂. The red solid line and filled diamond show the measurement by BBCES and CRDS, respectively, with black dashed line showing the power-law fit to the measured data. Several other literature parameterizations are shown in solid colored lines, and discussed further in the text. The ratio of each trace relative to the power-law fit derived here is shown at the bottom, and all are within 10% of the fit derived here. The small peaks and valleys observed in**
**the experimental fit are due to imperfect subtraction of peaks from O₂ and the O₄ oxygen dimer.**



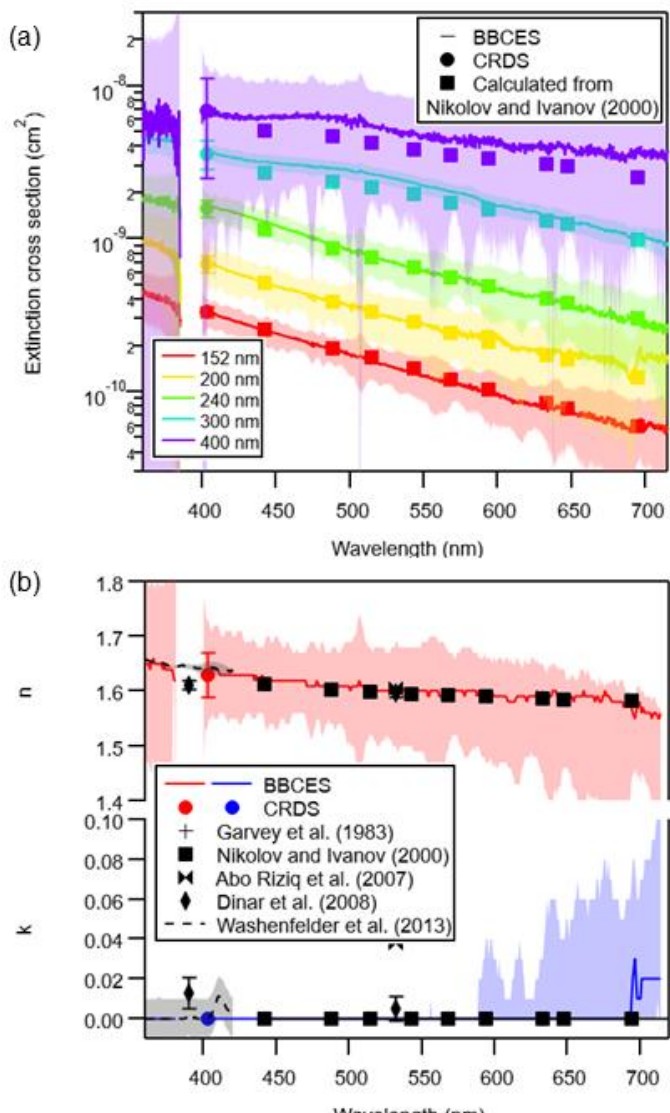

**Figure 3. (a)** Measured extinction cross sections for five diameters of PSL spheres are shown as solid lines, with $2\sigma$ error bars designated by shaded regions. The theoretical extinction cross section calculated from the RI of Nikolov and Ivanov (2000) and the measured aerosol size distribution is shown as filled squares for each diameter set point. **(b)** The retrieved RI of PSL spheres derived from the measurement in (a) and the size distribution measured by the OPC. The RI is retrieved to the nearest 0.01, and the absence of a visible upper or lower error bar implies the calculated error is less than 0.01. The RI retrieved by Nikolov an Ivanov (2000) is shown in black squares. The error increases at longer wavelengths due to increased uncertainty in the $CO_2$ Rayleigh scattering spectrum.





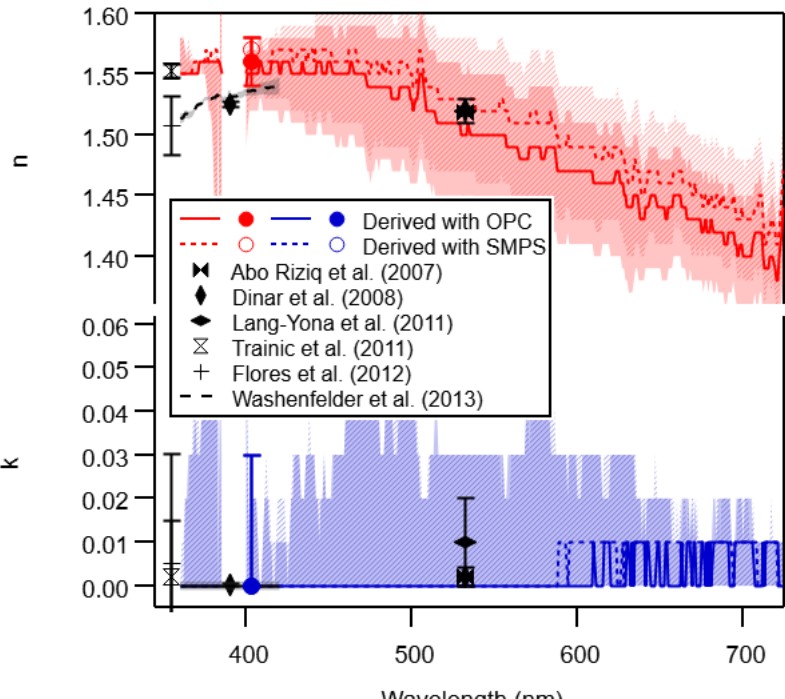


**Figure 4. The RI retrieved for ammonium sulfate aerosol, using two particle size distribution methods: an OPC (LAS 3340) and a DMA operating in SMPS mode. Both retrievals agree well with literature values for ammonium sulfate between 350 and 550 nm. As in Fig 3, the RI is retrieved to the nearest 0.01, and an absence of a visible error bar implies that the calculated error is smaller than 0.01.**





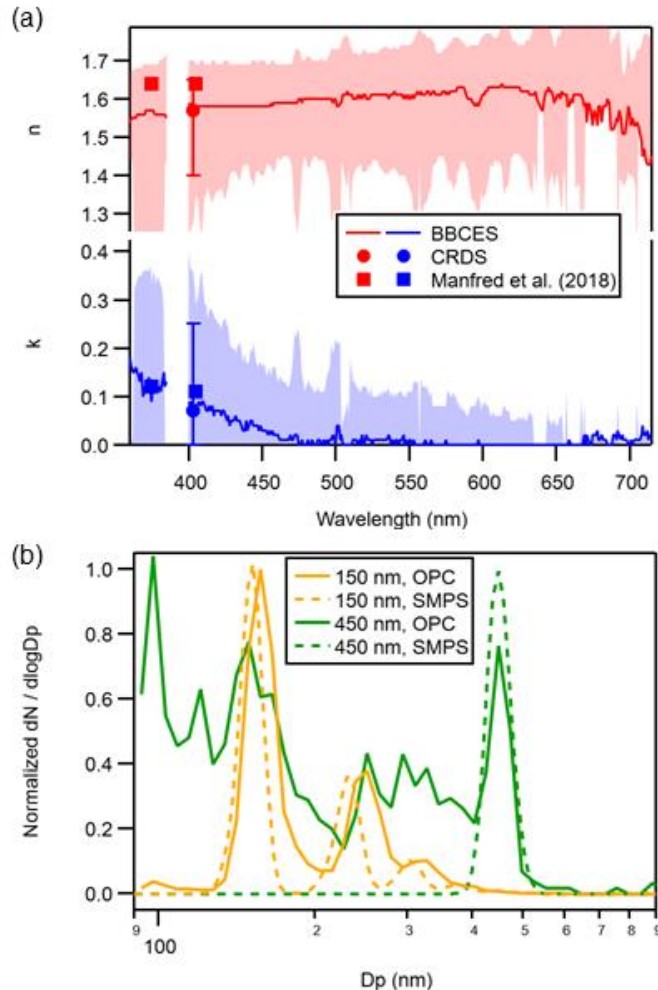


**Figure 5. (a) Retrieved RI of undenuded smoke during Fire A, a primarily brown carbon fire. The BBCES and CRDS retrievals are shown in solid line and circular markers. The squares indicate the assumed RI in Manfred et al. (2018) for this fire. The increase in the scattering component, $k$, with decreasing wavelength is characteristic of brown carbon aerosol. (b) The normalized size distribution of the aerosol from the OPC and the SMPS for two size set points, 150 and 450 nm. The 150 nm OPC measurement has a small peak near 100 nm but otherwise shows reasonable agreement with the SMPS, while the 450 nm setpoint OPC distribution shows a significant fraction of the aerosol is smaller than 450 nm, which is not reflected in the SMPS estimate. The other size setpoints lie in between these two extremes.**




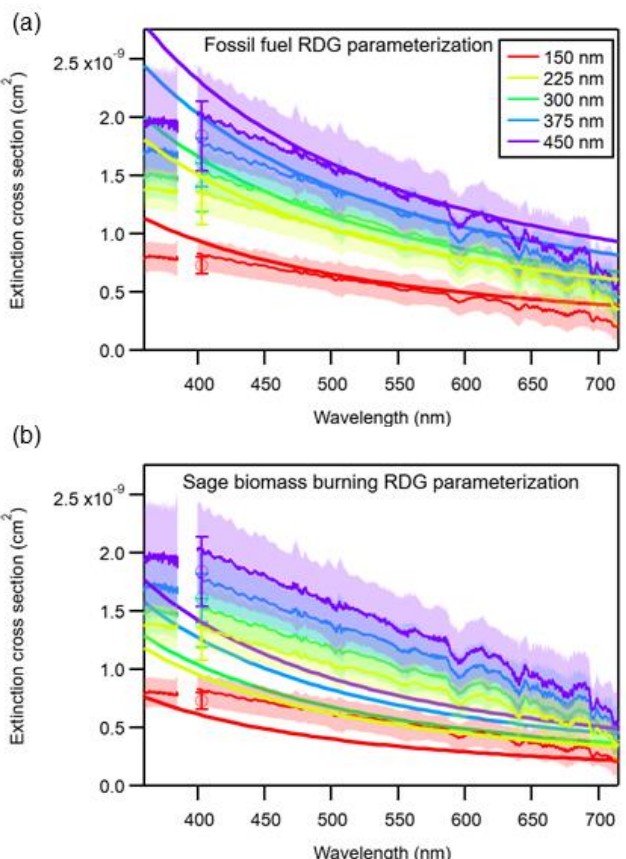

**Figure 6.** Measured extinction cross section of thermodenuded black carbon dominated aerosol in Fire B are shown in the thin solid lines. Calculated extinction cross section for different RDG parameters (Eq. (4)) are shown in each panel. (a) Fossil fuel RDG parameters from Sorensen et al (2001); (b) Biomass burning RDG parameters for sage from Chakrabarty (2006). The measurement lies between these two parameterizations, thereby constraining the RDG prefactor and fractal dimension, but is closer to the fossil fuel parameterization.





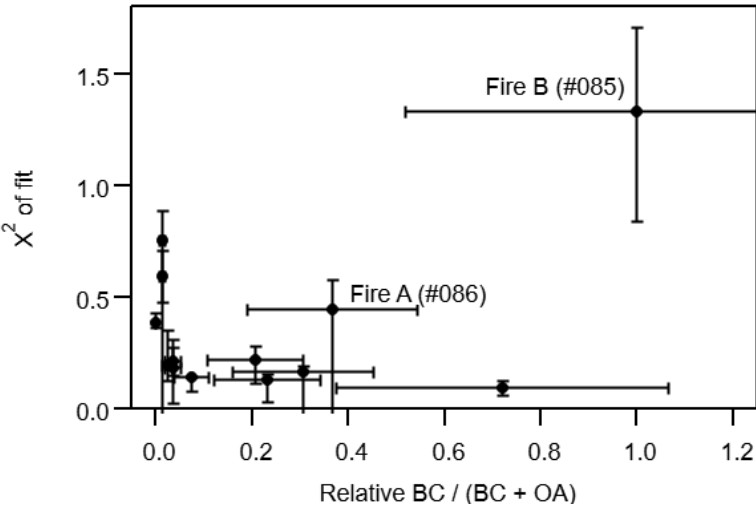


**Figure 7. An analysis of thirteen fires at the Fire Lab, including the two presented in detail in this paper. Each fire was analysed identically in this figure. For each fire, the $\chi^2$ of the retrieval fit assuming Mie theory is plotted against the relative fraction of total aerosol that is BC. There is a correlation between the BC fraction and the quality of the fit assuming Mie theory. Fire B has a far higher fraction of BC than the other fires and is fit very poorly assuming Mie theory, while Fire A is fit more accurately by Mie**

**theory. The majority of fires have a smaller BC contribution than either Fire A or B, and are generally fit reasonably well assuming Mie theory. Note that we have plotted relative BC fraction rather than absolute fraction. Significant uncertainties in the AMS data make the absolute fraction difficult to accurately assess, but the relative fraction is still robust. Therefore, Fire B, which had the highest BC fraction, is defined here as 1, and the BC fractions for the other fires are scaled relative to Fire B.**



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
