# Peer review of "Complex refractive indices in the ultraviolet and visible spectral region for highly absorbing non-spherical biomass burning aerosol"

_Atmospheric Chemistry and Physics, 2020_

## Referee Comment (RC1) · Anonymous Referee #1 · 5 Jan 2021

This manuscript communicates a new optical instrument that helps address the ever-important need to derive complex refractive indices from biomass burning aerosol, a key parameter in satellite retrieval algorithms and climate modeling. The authors also describe its deployment to the 2016 FIREX experiments and the results from several biomass burning experiments. It is unsurprising that light scattering from fractal BC aggregates cannot be computed from Mie theory, which follows immediately from the requirement of spherical particles. It is even less surprising that the authors conclude that the retrievals are quite sensitive to the size distribution measurement, as errors in broad size distribution measurements will likely dominate the errors propagated from their optical instruments.

[Figure]

I recommend that this manuscript be published pending some minor technical corrections and the satisfactory response to a few minor questions and comments I have below. This manuscript is quite well-written, clearly organized, and follows a logical path from introduction to conclusion. I thank the authors for the opportunity to review their work.

Comments and questions

I appreciate the discussion of the difficulty in ascertaining the error in refractive index, this is something that the community seems to be in disagreement over, though this leads me to a larger point of how we discuss the refractive index in aerosol optics. Fundamentally, it is inappropriate to report the refractive index as m = (n $\pm$ $\sigma$n) + (k $\pm$ $\sigma$k), since n and k aren't independent of one another, and are rather functions of each other through the Kramers-Kronig relations. The refractive index is a single, complex quantity. However, given the way you conducted your experiments, I believe the way you represented your errors is appropriate. However, in section 4.3, you refer to n as the "scattering component" and k as the "absorbing component". Both n and k contribute to the overall optical behavior. I suggest re-writing this section to remove the notion that n and k are purely responsible for scattering and absorption, respectively. Using "real part" and "imaginary part" is appropriate.

How fast was your SMPS scan? I'm curious about the width of the transfer function.

In the second paragraph in section 2.5, you assert that aerosol RI is unaffected by wall losses. While this may be practically true, this is not universally true, especially for the reason you state. It certainly is for a homogeneous substance such as pure ammonium sulfate aerosol or PSL spheres. For a complex mixture such as biomass burning, aerosol RI is rather the "effective aerosol RI", where the particle behaves as if it has a single RI. In turn, the entire aerosol population being sampled will behave as if it has a single effective RI. However, getting down to the losses of particles of certain sizes, I would expect that the volatility of different compounds may preferentially distribute

them to smaller or larger particles, and therefore losses above or below a particular size range may impact your measurements. I would certainly expect the effect to be slight, and it in fact may be negligible within the accuracy of your measurements. If this is the case, this is no more than an extremely minor semantics quibble, however, I believe this section would benefit from a more accurate statement of exactly why you assume wall losses to be negligible on the optical properties of the population as a whole. You do address this in the first paragraph of section 3.3 where you state that you assume the RI does not vary systematically with size. This statement largely satisfies this comment.

Minor comments and technical corrections

Line 104 – "The very broadband..." language is vague and subjective. Consider removing "very". Line 277 – suggest changing "2$\times$ and 3$\times$ greater" to "two and three times greater". Line 284 – typo, change "existing" to "exiting". Line 297 – Consider moving "Theoretically" to the beginning of the sentence: "Theoretical particle losses between the DMA..." In section 3.3.2, you state Mie theory is valid when the size parameter x is approximately 1. Mie theory is always valid, despite the size parameter. At lower size parameters, the Rayleigh approximation is perfectly fine, and at higher size parameters, geometric optics is a more useful approximation, since the number of terms you need for an accurate Mie theory calculation grows as x + 4x1/3 + 2 (Wiscombe 1980). Furthermore, B&H's original FORTRAN algorithm is only valid up to x $\approx$ 104 (Wolf and Voshchinnikov 2004). Since you use an adaptation of this code, I would rather state "The Mie theory algorithm we used here is valid when x $\approx$ 1." This is a minor suggestion, and the authors may take or leave it. Line 304 – "It is a series approximation that allows a mathematical representation of light with spheres..." is slightly awkward usage. Consider "It is a truncated infinite series representation of the electromagnetic field scattered from spheres..." or something similar. Line 411 – Remove comma after "function". Throughout – when the imaginary component of m is zero, write "0.00i" to be consistent with your measurement and retrieval precision.

Figure comments

In general, the figures are excellent. The instrument diagrams are fully informative without being cluttered, and I do appreciate that. The graphs, however, are coming across in the pdf as quite low resolution upon zooming in. This may be an artefact of the proof pdf, but do make sure that the journal has access to the highest quality figures you can produce. I would also like to caution that in some figures (notably 3b and 4), the legends dominate the figure area and distract from objective interpretation of the data. Consider moving the legends outside the figures. In figure 6, it is quite difficult to take in all that data at once. Is it necessary to have the calculated cross sections in each panel? If so, you have ten traces to keep track of. I suggest making the figure span the whole page and see if a logarithmic y-axis helps separate the data. As it is, it's extremely cluttered. For all figures where you present k, consider a logarithmic scale. Since k is highly sensitive, far more so than n, a logarithmic scale will better convey the spectral functionality at lower values. Finally, carefully consider the color schemes you use for data-dense graphs. When printed in grayscale or proofed for colorblindness, many traces are indistinguishable from one another.

References Wiscombe, W. J., Improved Mie scattering algorithms. Appl. Optics. 1980, 19 (9), 1505. Wolf, S., and Voshchinnikov, N. V., Mie Scattering by Ensembles of Particles with Very Large Size Parameters. Comput. Physics Commun. 2004, 162 (2), 113-123.

Please also note the supplement to this comment:
https://acp.copernicus.org/preprints/acp-2020-1200/acp-2020-1200-RC1-supplement.pdf

---

## Referee Comment (RC2) · Anonymous Referee #2 · 2 Feb 2021

The manuscript by Caroline et al. presents broadband optical measurements of biomass burning aerosol and additional scattering standards. These measurements and analysis are non-trivial, but this work shows that reducing the refractive index revival uncertainty will become critical for ambient measurements and pushing for improvements in global models.

The manuscript is written clearly and added a significant contribution to the community.

Comments:

Figure 4 and related discussion: There is an additional literature line to add from: Bain, A., Rafferty, A., and Preston, T. C.: The Wavelength Dependent Complex Refractive Index of Hygroscopic Aerosol Particles and Other Aqueous Media: An Effective Oscillator Model, Geophys. Res. Lett., 46(17–18), 10636–10645, doi:10.1029/2019GL084568, 2019. Note, the Bain et al. paper was for aqueous solutions, so depending on your humidity, this may not be a valid comparison.

The Bain et al. paper also should be brought into the new discussion of the Kramers-Kronig relation suggested by the other reviewer.

Figure 7 and other burns: Given you are reporting on 13 burns completed and show fit quality in Figure 7, It would be informative to add the refractive index spectra for the other burns in the supplemental information. A repeat of Figure 5a for each burn in the SI.

Line 24: The spacing is inconsistent in the reported refractive index. Also, I think 1.635 ($\pm$ 0.056) + 0.6 ($\pm$ 0.056)i would read better, but that is your choice (or typesetter's).

---

## Author Comment (AC1) · 19 Mar 2021

The comment was uploaded in the form of a supplement:
https://acp.copernicus.org/preprints/acp-2020-1200/acp-2020-1200-AC1-supplement.pdf

---

## Author Response (AR2)

Response to Reviewers
Manuscript Number: ACP-2020-1200
Manuscript Title: Complex refractive indices in the ultraviolet and visible spectral region for highly absorbing non-spherical biomass burning aerosol

The discussion below includes the complete text from the reviewer (in **bold**), along with our responses to the specific comments (in normal text) and the corresponding changes (in red) made to the revised manuscript. All of the line numbers refer to the revised manuscript with tracked changes.

**Response to Reviewer #1 Comments:**

**This manuscript communicates a new optical instrument that helps address the ever-important need to derive complex refractive indices from biomass burning aerosol, a key parameter in satellite retrieval algorithms and climate modeling. The authors also describe its deployment to the 2016 FIREX experiments and the results from several biomass burning experiments. It is unsurprising that light scattering from fractal BC aggregates cannot be computed from Mie theory, which follows immediately from the requirement of spherical particles. It is even less surprising that the authors conclude that the retrievals are quite sensitive to the size distribution measurement, as errors in broad size distribution measurements will likely dominate the errors propagated from their optical instruments.**

**I recommend that this manuscript be published pending some minor technical corrections and the satisfactory response to a few minor questions and comments I have below. This manuscript is quite well-written, clearly organized, and follows a logical path from introduction to conclusion. I thank the authors for the opportunity to review their work.**

We thank the reviewer for the positive review and helpful comments. Listed below are our responses to the comments and the corresponding changes made to the revised manuscript.

**I appreciate the discussion of the difficulty in ascertaining the error in refractive index, this is something that the community seems to be in disagreement over, though this leads me to a larger point of how we discuss the refractive index in aerosol optics. Fundamentally, it is inappropriate to report the refractive index as m = ($n \pm \sigma_n$) + ($k \pm \sigma_k$), since n and k aren't independent of one another, and are rather functions of each other through the Kramers-Kronig relations. The refractive index is a single, complex quantity. However, given the way you conducted your experiments, I believe the way you represented your errors is appropriate. However, in section 4.3, you refer to n as the "scattering component" and k as the "absorbing component". Both n and k contribute to the overall optical behavior. I suggest re-writing this section to remove the notion that n and k are purely responsible for scattering and absorption, respectively. Using "real part" and "imaginary part" is appropriate.**

We thank the reviewer for this comment. We have changed any text that implies that the complex refractive index can be separated into scattering and absorbing components. Specifically:

Lines 39-40: The RI is an intrinsic physical property of the particle, and is described as m = n + ki (Moosmüller et al., 2009; Moise et al., 2015).

Lines 269-270: Since the RI consists of , *both* n and k, at least two extinction measurements are required to retrieve these two  *parameters* (Bluvshtein et 265 al., 2012).

Lines 404-406: We find that  n *varies from* 1.65 to 1.57 between 360 and 700 nm, while  k = 0 throughout, as expected for purely scattering particles.

Lines 450-451: If  these brown carbon particles *scatter light more effectively than ammonium sulfate particles* , then the OPC will interpret this increased light scattering as a larger particle.

Lines 609-610: The increase in  k with decreasing wavelength is characteristic of brown carbon aerosol.

**How fast was your SMPS scan? I'm curious about the width of the transfer function.**

The total SMPS scan time was 240 s (110 s upscan, 20 s held at 5000V, and 110 s downscan). We have added this information to the text:

Lines 135-138: At regular intervals, the DMA and CPC were operated as a scanning mobility particle sizer (SMPS) to determine the aerosol size distribution by scanning the DMA column voltage  *up and then down between* 0 – 5000 V *over 240 s*, and applying an inversion algorithm (Twomey, 1975; Markowski, 1987).

**In the second paragraph in section 2.5, you assert that aerosol RI is unaffected by wall losses. While this may be practically true, this is not universally true, especially for the reason you state. It certainly is for a homogeneous substance such as pure ammonium sulfate aerosol or PSL spheres. For a complex mixture such as biomass burning, aerosol RI is rather the "effective aerosol RI", where the particle behaves as if it has a single RI. In turn, the entire aerosol population being sampled will behave as if it has a single effective RI. However, getting down to the losses of particles of certain sizes, I would expect that the volatility of different compounds may preferentially distribute them to smaller or larger particles, and therefore losses above or below a particular size range may impact your measurements. I would certainly expect the effect to be slight, and it in fact may be negligible within the accuracy of your measurements. If this is the case, this is no more than an extremely minor semantics quibble, however, I believe this section would benefit from a more accurate statement of exactly why you assume wall losses to be negligible on the optical properties of the population as a whole. You do address this in the first paragraph of section 3.3 where you state that you assume the RI does not vary systematically with size. This statement largely satisfies this comment.**

We thank the reviewer for this comment. It is true that wall losses in the thermodenuder may affect aerosol of different size differently. However, as the reviewer points out, this analysis necessarily measures an effective RI that is assumed to be size-independent. Therefore, we don't expect these losses to affect our results. We have added more text to state this more clearly, along with a reference to Moise (2015).

Lines 183-186: The throughput efficiency was found to be less than unity (86 ± 4% for particles between 100 and 300 nm) due to thermophoretic wall losses, but the analysis of the aerosol *size-independent* RI, which is an intrinsic property of the aerosol *(Moise et al. 2015)*, is *largely* unaffected by these losses. *We therefore do not make any corrections to these data.*

**Line 106 – "The very broadband…" language is vague and subjective. Consider removing "very".**

Corrected.

**Line 283 – suggest changing "2× and 3× greater" to "two and three times greater".**

Corrected.

**Line 289 – typo, change "existing" to "exiting".**

Corrected.

**Line 303 – Consider moving "Theoretically" to the beginning of the sentence: "Theoretical particle losses between the DMA…"**

Corrected.

**In Section 3.3.2, you state Mie theory is valid when the size parameter x is approximately 1. Mie theory is always valid, despite the size parameter. At lower size parameters, the Rayleigh approximation is perfectly fine, and at higher size parameters, geometric optics is a more useful approximation, since the number of terms you need for an accurate Mie theory calculation grows as $x + 4x^{1/3} + 2$ (Wiscombe 1980). Furthermore, B&H's original FORTRAN algorithm is only valid up to $x \approx 104$ (Wolf and Voshchinnikov 2004). Since you use an adaptation of this code, I would rather state "The Mie theory algorithm we used here is valid when $x \approx 1$." This is a minor suggestion, and the authors may take or leave it.**

We have edited the text to correct this:

Lines 309-313: "Mie theory is a solution to Maxwell's equations that describes the interaction of light with homogeneous, spherical particles, when the diameter of the sphere is similar to the wavelength of light (Bohren and Huffman, 1983). It is a series approximation that allows a mathematical representation of light with spheres, concentric spheres, and clusters of spheres. *For the representation used in this work, the theory is valid when the dimensionless size parameter (x = πd / λ) is approximately 1.*

**Line 311 – "It is a series approximation that allows a mathematical representation of light with spheres…" is slightly awkward usage. Consider "It is a truncated infinite series representation of the electromagnetic field scattered from spheres…" or something similar.**

We have changed this:

Line 311:  *It is a truncated infinite series representing the electromagnetic field scattered from spheres.*

**Line 411 – Remove comma after "function".**

Corrected.

**Throughout – when the imaginary component of m is zero, write "0.00i" to be consistent with your measurement and retrieval precision.**

We thank the reviewer for this comment, and have changed the text from 0*i* to 0.00*i* in Lines 408, 421, 450, and 516.

**In general, the figures are excellent. The instrument diagrams are fully informative without being cluttered, and I do appreciate that. The graphs, however, are coming across in the pdf as quite low resolution upon zooming in. This may be an artefact of the proof pdf, but do make sure that the journal has access to the highest quality figures you can produce. I would also like to caution that in some figures (notably 3b and 4), the legends dominate the figure area and distract from objective interpretation of the data. Consider moving the legends outside the figures.**

We will confirm that the final manuscript has high resolution figures. We have also moved the legends in figures 3b and 4 to be just above the figure, to make the graphs easier to read.

**In figure 6, it is quite difficult to take in all that data at once. Is it necessary to have the calculated cross sections in each panel? If so, you have ten traces to keep track of. I suggest making the**

**figure span the whole page and see if a logarithmic y-axis helps separate the data. As it is, it's extremely cluttered.**

We have attempted to make the figure less cluttered and easier to read by expanding the vertical height, and changing the theoretically calculated traces to dashed lines. Unfortunately, a logarithmic y-axis did not help the clarity of the figure, so we have retained the linear axis scale. Additionally, because panels a and b are comparing two different model parameterizations (dashed line) to the observations (solid line), we do feel it is necessary to include the observations in both panels, to make the comparison easy to see.

**For all figures where you present k, consider a logarithmic scale. Since k is highly sensitive, far more so than n, a logarithmic scale will better convey the spectral functionality at lower values.**

We agree with the reviewer that k is a highly sensitive parameter. However, in this analysis, we determine k to the nearest 0.01. Therefore, any k that is less than 0.005 is rounded down to 0, which will cause issues on a logarithmic scale. To clarify this, we emphasize that k is calculated to the nearest 0.01 in the captions of figure 5, as well as in figures 3 and 4.

Lines 608 - 609: The BBCES and CRDS retrievals are shown in solid line and circular markers, *and are calculated to the nearest 0.01.*

**Finally, carefully consider the color schemes you use for data-dense graphs. When printed in grayscale or proofed for colorblindness, many traces are indistinguishable from one another.**

We thank the reviewer for pointing this out. We have changed the color scheme for all the color figures to one recommended for colorblindness in Figure 2 of Wong et al (2011). We then used the Color Oracle tool (colororacle.org) to ensure that all traces are distinguishable for the most common types of colorblindness (Deuteranopia and Protanopia).

References
Wiscombe, W. J., Improved Mie scattering algorithms. Appl. Optics. 1980, 19 (9), 1505.

Wolf, S., and Voshchinnikov, N. V., Mie Scattering by Ensembles of Particles with Very Large Size Parameters. Comput. Physics Commun. 2004, 162 (2), 113-123.

Wong, B., Points of View: Color blindness, Nature Methods, 2011, 8, 441.

**Response to Reviewer #2 Comments:**
* * *
**The manuscript by Caroline et al. presents broadband optical measurements of biomass burning aerosol and additional scattering standards. These measurements and analysis are non-trivial, but this work shows that reducing the refractive index revival uncertainty will become critical for ambient measurements and pushing for improvements in global models.**

**The manuscript is written clearly and added a significant contribution to the community.**

We thank the reviewer for the positive review. Listed below are our responses to the comments and the corresponding changes made to the revised manuscript.

**Figure 4 and related discussion: There is an additional literature line to add from: Bain, A., Rafferty, A., and Preston, T. C.: The Wavelength Dependent Complex Refractive Index of Hygroscopic Aerosol Particles and Other Aqueous Media: An Effective Oscillator Model, Geophys. Res. Lett., 46(17–18), 10636–10645, doi:10.1029/2019GL084568, 2019. Note, the Bain et al. paper was for aqueous solutions, so depending on your humidity, this may not be a valid comparison.**

The complex refractive indices shown in Figure 4 represent dry ammonium sulfate aerosol (RH < 5%). This is described in the main text. We have changed the caption to clarify this:

Line 603: "Figure 4. The RI retrieved for *dry* ammonium sulfate aerosol, using two particle size distribution methods…"

The complex refractive indices reported in Bain et al. 2019 are for aqueous solutions of ammonium sulfate, with a minimum water activity of 0.3, and unfortunately cannot be directly compared to the values in Figure 4.

**The Bain et al. paper also should be brought into the new discussion of the Kramers-Kronig relation suggested by the other reviewer.**

Because n and k are related through the Kramers-Kronig relations, we have changed the text to be clear that the complex refractive index cannot be separated into scattering and absorbing components. Specifically:

Lines 39-40: The RI is an intrinsic physical property of the particle, and is described as m = n + ki, where n represents the scattering component and k represents the absorbing component (Moosmüller et al., 2009; Moise et al., 2015).

Lines 269-270: Since the RI consists of a scattering and an absorbing component, *both* n and k, at least two extinction measurements are required to retrieve these two variables (Bluvshtein et 265 al., 2012).

Lines 404-406: We find that the scattering component varies from n *varies from* 1.65 to 1.57 between 360 and 700 nm, while the absorbing component is constant at k = 0 throughout, as expected for purely scattering particles.

Lines 450-451: If the RI of these brown carbon particles *scatter light more effectively than ammonium sulfate particles* has greater scattering component in the RI than 1.52, then the OPC will interpret this increased light scattering as a larger particle.

Lines 609-610: The increase in the scattering component, k, with decreasing wavelength is characteristic of brown carbon aerosol.

**Figure 7 and other burns: Given you are reporting on 13 burns completed and show fit quality in Figure 7, It would be informative to add the refractive index spectra for the other burns in the supplemental information. A repeat of Figure 5a for each burn in the SI.**

We thank the reviewer for this suggestion, and have prepared a new figure for the SI (Fig. S6) that shows the complex refractive index values for each fire, similar to Figure 5a. We elected to plot all 13 fires together, rather than make 13 individual plots, to demonstrate the similarity between many of the fires, and the non-physical behavior of Fire B. We have also included average values for n and k at 365 nm, as this is a wavelength that other researchers may find useful

Lines 511 - 513: *Figure S6 shows the retrieved wavelength-dependent RI for each fire across the entire 360 – 720 nm wavelength range. To demonstrate the effectiveness of the retrieval method,* we report the $\chi^2$ of the RI retrieval in *Fig. 7* at *a single wavelength,* 475 nm, where the instrument had high mirror reflectivity and therefore good precision.

Line 24-25: *and at 365 nm, the average refractive index is 1.605 (± 0.041) + 0.038 (± 0.074)i.*

Line 517: *At 365 nm, the average value of the real part was 1.605 ± 0.041 and the imaginary part was 0.038i ± 0.074.*

**Line 24: The spacing is inconsistent in the reported refractive index. Also, I think 1.635 (± 0.056) + 0.6 (± 0.056)i would read better, but that is your choice (or typesetter's).**

We have corrected the spacing and changed the format:

Line 24: (1.635 ± 0.056) + (0.06 ±0.12)i changed to *1.635 (± 0.056) + 0.06 (± 0.12)i*.

**Other changes:**

Following input from our coauthors, we have made several other minor changes to the manuscript:

We have included Nicholas Wagner's new affiliation as:

Line 10: *ˣNow at Ball Aerospace, Broomfield, CO 80021, USA*

To clarify that the LAS is an example of an OPC, we have adjusted the following sentence:

Line 141: The optical particle counter (OPC*) used here was a laser aerosol spectrometer* (LAS 3340, TSI Inc., Shoreview, MN, USA), *which* detects light scattered by…

We have clarified the causes behind the increase uncertainty in the AMS measurements:

Line 197: The uncertainty of the organic aerosol mass is typically ~38% *due to standard uncertainties in AMS measurements* (Bahreini et al. 2009), *but was greater for biomass burning aerosol at the Fire Sciences Laboratory because of additional variability in aerosol volatility due to dilution of the dense smoke.*

We have included a reference to Adler (2019) when discussing non-spherical particles transmitted through the DMA:

Line 293-294: …the transmission through the DMA is affected by the aerodynamic resistance of the sheath flow *(Adler et al. 2019)*.

We have added an additional statement that the freshly-emitted smoke from the Fire Sciences laboratory may not reflect real smoke's contribution from dust and biological fragments:

Line 512: We have added the following line to address the editor's suggestion to address the fact that the averaged k value is slightly larger at 475nm than at 365nm. *We note that the average k value is slightly greater at 475 nm than at 365 nm, but the two values are consistent within the error bars of the averaged measurements of aerosol from different fuel types and fire conditions.*

Line 525-528: However, we note that this freshly-emitted smoke was from small scale burns under controlled conditions, and may not represent the full range of smoke aerosol types in ambient aerosol, *including dust and biological fragments. Additionally,* further downwind of the fire, coagulation and fractal aggregate collapse can alter the particle morphology.

We have clarified the first sentence of the Conclusions section, to better characterize the manuscript:

Line 535-536: This paper describes the development of a new broadband cavity enhanced spectrometer that *derives the RI of biomass burning aerosol with low BC content*  over a very broadband wavelength region.
We have included the Cooperative Agreement in the list of acknowledgements:

Line 575: … *and the NOAA Cooperative Agreement with CIRES, NA17OAR4320101.*